# A Novel ANN-PSO Method for Optimizing a Small-Signal Equivalent Model of a Dual-Field-Plate GaN HEMT

**DOI:** 10.3390/mi15121437

**Published:** 2024-11-28

**Authors:** Haowen Shen, Wenyong Zhou, Jinye Wang, Hangjiang Jin, Yifan Wu, Junchao Wang, Jun Liu

**Affiliations:** 1Innovation Center for Electronic Design Automation Technology, Hangzhou Dianzi University, Hangzhou 310018, China; hwshen@hdu.edu.cn (H.S.);; 2Empyrean Technology Co., Ltd., Beijing 100102, China

**Keywords:** GaN HEMT, equivalent circuit modeling, ANN, PSO, parameter optimization

## Abstract

This study introduces a novel method that integrates artificial neural networks (ANNs) with the Particle Swarm Optimization (PSO) algorithm to enhance the efficiency and precision of parameter optimization for the small-signal equivalent model of dual-field-plate GaN HEMT devices. We initially train an ANN model to predict the S-parameters of the device, and subsequently utilize the PSO algorithm for parameter optimization. Comparative analysis with the NSGA2 and DE algorithms, based on convergence speed and accuracy, underscores the superiority of the PSO algorithm. Ultimately, this ANN-PSO approach is employed to automatically optimize the internal parameters of a 4 × 250 μm dual-field-plate GaN HEMT equivalent circuit model within the frequency range of 1–18 GHz. The method’s effectiveness under varying bias conditions is validated through comparison with traditional physical formula analysis methods. The results demonstrate that the ANN-PSO method significantly enhances the automation and efficiency of parameter optimization while maintaining model accuracy, providing a reference for the optimization of other device models.

## 1. Introduction

GaN HEMTs hold significant potential for development in the RF domain [1,2,3] due to their remarkable properties such as high electron mobility, saturation velocity, and thermal stability [4]. However, GaN HEMT devices still encounter challenges like breakdown and leakage currents under high-electric-field and power operation conditions. The field plate (FP) is a widely adopted electric field optimization technology [5], and various types of GaN HEMT devices [6,7,8] featuring field-plate structures have been proposed and fabricated. Research has demonstrated that GaN HEMTs, after applying an FP, can effectively balance the internal electric field of the device [9,10], significantly improve the breakdown voltage [11], and suppress current collapse [12,13]. Notably, devices with dual field plates (DFPs), integrating both a source field plate (SFP) and a gate field plate (GFP), exhibit higher breakdown voltage [14] and lower dynamic on-resistance [15] compared to those with a single FP.

As the manufacturing of GaN HEMTs with FPs becomes increasingly prevalent, the conventional small-signal equivalent model for GaN HEMTs [16,17,18] proves increasingly inadequate for accurately depicting their physical behavior. Substantial research has been dedicated to modeling FPs in GaN HEMTs [19,20,21]. Notably, the research conducted by JY Wang et al. [22] introduced an innovative small-signal model for GaN HEMTs by incorporating an equivalent circuit for the FP. While this advancement significantly enhances the model’s accuracy, it also increases the complexity of the equivalent circuit. Owing to the reliance on intricate physical formula derivation and the extensive knowledge possessed by engineers, the time required to extract and optimize the parameters of the new small-signal model is substantially greater than that for traditional models.

To enhance the efficiency of optimizing this novel small-signal equivalent model, we propose utilizing an ANN as a surrogate for traditional simulators. Neural networks are well regarded in the realm of RF and microwave modeling and design [23] due to their robust learning capabilities in capturing nonlinear input–output relationships. There is a wide range of research on GaN technology using AI applications [24,25,26,27]; in the context of GaN HEMTs, extensive research has employed neural networks to predict device breakdown characteristics [28], current–voltage (I–V) characteristics [29], and small-signal (S-parameter) characteristics [30,31,32]. However, the majority of these neural network training studies are based on three-dimensional electromagnetic field simulations in TCAD or other software, with relatively fewer studies focusing on SPICE simulations, particularly those involving DFP equivalent structures.

In the domain of model parameter extraction algorithms, the majority of research gravitates towards evolutionary algorithms, including popular ones such as genetic algorithms [33,34,35] and PSO [36,37]. A minority of researchers have proposed approaches that integrate algorithms with neural networks [38], achieving commendable results. However, there is still a significant gap in research that effectively marries well-trained neural network models with optimization techniques to further enhance the optimization of GaN HEMT model parameters.

This paper initially introduces the DFP GaN HEMT device and its small-signal equivalent model. It then presents the ANN model trained for this model. Finally, it proposes a method that applies the ANN model to the PSO algorithm to optimize the small-signal model parameters of the DFP GaN HEMT device equivalent circuit and contrasts it with the precision and time expenditure of models constructed through conventional analytical methods. Verification is conducted on a DFP GaN HEMT fabricated using a 450 nm process, featuring four gate fingers, each measuring 250 μm. The results demonstrate that, compared to the traditional method of parameter optimization using physical formula analysis, this approach not only ensures model accuracy but also automates parameter optimization, thereby enhancing efficiency.

## 2. Materials and Methods

### 2.1. Novel Device Model

Figure 1 illustrates a cross-section of a DFP GaN HEMT manufactured in China, which integrates a GFP and an SFP to simultaneously enhance collapse, breakdown, and gain characteristics [22]. The gate length ($L_{g}$) is 0.45 μm, the GFP length ($L_{gfp}$) is 0.3 μm, and the SFP length ($L_{sfp}$) is 1.2 μm. The length of the source access domain ($L_{gs}$) is 1 μm, and the length of the drain access domain ($L_{gd}$) is 3 μm. These characteristic dimensions are sourced from Reference [22] and are typically provided by the manufacturer upon device delivery, as SPICE model device modeling often follows actual production. Our modeling was executed with precise knowledge of the device’s characteristic dimensions. Compared to the conventional GaN HEMT device model, this model exhibits greater topological complexity, and the analytical effort required to derive reasonable parameters using traditional methods is excessively high. Thus, we chose this model to demonstrate and validate our work.

We selected 450 nm since we have already conducted tests on devices of this size using the Keysight N5247A PNA-X produced by Keysight Technologies microwave network analyzer, which provides more accurate data than theoretical predictions. The measurement setup was calibrated using the short–open–load–through (SOLT) method. It is noteworthy that the method proposed in this paper is based on modifications made to the equivalent structure of the DFP HEMT. This equivalent structure is not directly related to the device’s dimensions. In fact, the gate length does not directly influence the internal parameter extraction throughout the automated optimization process. As long as it is within a reasonable range for manufacturable devices, this method can be applied.

The latest small-signal equivalent model structure of the device [22] is depicted in Figure 2, which is segmented into four principal parts. The area outside the dashed line signifies the peripheral equivalent, the area within the red box denotes the equivalent subcircuit of the intrinsic transistor, and the blue and green boxes represent the equivalent subcircuits of the GFP and the SFP, respectively. This equivalent structure applies to all GFP devices. Consequently, based on this equivalent topological structure, the optimization methods proposed in our work are not limited to the devices in Reference [22] but are universally applicable to all FP devices.

After introducing the two field plates, the intrinsic parameters of the novel HEMT equivalent circuit model reached 14 ($C_{gs1}$, $C_{gd1}$, $C_{ds1}$, $C_{gs2}$, $C_{gd2}$, $C_{ds2}$, $C_{ds3}$, $R_{ds1}$, $R_{ds2}$, $R_{ds3}$, $g_{m1}$, $g_{m2}$, $\tau_{1}$, $\tau_{2}$), these model parameters will be continuously optimized to adjust the model accuracy.

### 2.2. Model Optimization Method

The optimization method for the small-signal equivalent model of DFP GaN HEMT devices can be succinctly summarized as follows: training a neural network model and optimizing with PSO. As illustrated in Figure 3, the principal steps involved are producing datasets, training the ANN model, and invoking the PSO algorithm.

#### 2.2.1. Producing Datasets

The small-signal equivalent circuit utilized in this study is depicted in Figure 2. Notably, the parameters of the peripheral components ($C_{pda}$, $C_{gda}$, $C_{pga}$, $C_{pgi}$, $C_{pdi}$, $C_{gdi}$, $L_{g}$, $L_{d}$, $L_{s}$, $R_{g}$, $R_{d}$, $R_{s}$) remain independent of the bias. Modelers can determine these parameters under “zero drain bias pinched-off conditions” using low-frequency Y parameters [39]. The equivalent subcircuits within the three dashed boxes are influenced by the bias conditions of the device’s operation. We extricate these bias-dependent internal parameters from the circuit and employ them as input conditions for the trained neural network, with the simulated S-parameter values of the device model serving as the output. We opted to use the Advanced Design System (ADS) 2017 simulator from Keysight Technologies to sample the datasets. The range of input parameter values is detailed in Table 1, which corresponds to those set by engineers during optimization. We randomly sampled and simulated 10,000 sets of data under bias conditions of $V_{ds}$ = 28 V, 40 V, 48 V, corresponding to $V_{gs}$ = 1 V, −1 V, −3 V, which contained one set of on-state and two sets of off-state working conditions. These three conditions represent potential voltage scenarios encountered during HEMT operation. Most other bias conditions fall within the scope described by these three scenarios. In exceptional circumstances, one can adhere to the following procedure: initially, gather samples to construct the database; subsequently, train the ANN; and finally, employ PSO for optimization to derive model parameters. Throughout this sequence, the configurations can remain constant.

During model training, using a dataset with high similarity can significantly enhance training accuracy. To ensure the model’s high precision, we meticulously curated the datasets, excluding interference data under resonance and other anomalous conditions, as shown in Figure 4. We set a specific threshold (0.5 in Figure 4) to identify and exclude the interference curves that deviate significantly from the majority. This strategy helps filter out potential outliers that might affect model performance, thereby ensuring robustness and accuracy in the training results. Following this data refinement, 5245 datasets were retained for the bias condition of $V_{ds}$ = 28 V, $V_{gs}$ = 1 V, while the other two bias datasets consistently retained 10,000 sets each. These operations effectively mitigate the “garbage in, garbage out” issue in training and bolster the model’s reliability. Subsequent results indicated that after data cleaning, the model training accuracy increased from 78% to 99%.

We meticulously divided our dataset into an 80% training set and 20% test set. This ratio aims to balance the thoroughness of model training with the accuracy of validation. By utilizing the majority of the data for training, we ensure that the neural network can learn the complex patterns and relationships within the data. Meanwhile, the retained 20% test set provides an independent and representative sample set for the validation of model performance. This division strategy not only provides our model with ample training data to capture subtle features within the data but also ensures that we can accurately assess the model’s generalization capabilities through a significant test set. In this way, we can ensure that the model maintains a high level of performance and predictive accuracy when dealing with unseen samples.

#### 2.2.2. Training the ANN Model

After a meticulous data calibration process, our training set demonstrated significant similarity and consistency in the sample data, laying a solid foundation for building a robust neural network model. In light of this, we adopted the widely recognized hold-out method for model training and validation. This method is highly regarded for its simplicity and the absence of any prior assumptions about data distribution, ensuring that our research possessed high adaptability and universality.

During the training phase of the neural network, we normalized all data, which facilitated an accelerated convergence speed and enhanced both the model’s generalization capability and numerical stability. The loss function employed was the mean squared error (*MSE*), as illustrated in Formula (Equation 1). This loss function enabled smoother weight adjustments, mitigating significant weight changes caused by individual outliers. Formula (Equation 2) depicts the calculation method for the model’s R-squared (*R^2^*), a widely used metric to assess model precision. The closer its value is to 1, the higher the predictive accuracy of the model. We selected the Adam optimizer due to its ability to adaptively adjust the learning rate, thereby expediting the convergence process.
(1)$$MSE=\frac{1}{N}\sum_{i=1}^{N}{\left({\hat{y}}_{i}-y_{i}\right)}^{2}$$
(2)$$R^{2}=1-\frac{\sum_{i=1}^{N}{\left(y_{i}-{\hat{y}}_{i}\right)}^{2}}{\sum_{i=1}^{N}{\left(y_{i}-\bar{y}\right)}^{2}}$$

In Formulas (Equation 1) and (Equation 2), ${\hat{y}}_{i}$ is the predicted value of the model, while $\bar{y}$ denotes the mean value.

The final architecture of the ANN comprises 1 input layer, 2 hidden layers, and 1 output layer, as illustrated in Figure 5. The input layer contains 15 nodes, representing 14 internal parameters ($C_{gs1}$, $C_{gd1}$, $C_{ds1}$, $C_{gs2}$, $C_{gd2}$, $C_{ds2}$, $C_{ds3}$, $R_{ds1}$, $R_{ds2}$, $R_{ds3}$, $g_{m1}$, $g_{m2}$, $\tau_{1}$, $\tau_{2}$) and one frequency parameter (freq). The output layer comprises 8 nodes, each corresponding to the real and imaginary parts of $S_{11}$, $S_{12}$, $S_{21}$, $S_{22}$. Both hidden layers are composed of 256 neurons each. The ReLU activation function is utilized in the input and hidden layers to enhance the network’s nonlinear expression.

#### 2.2.3. Invoking the PSO Algorithm

The trained neural network model can function as a black-box model, synergized with other optimization algorithms. In this study, we employ the PSO algorithm in conjunction with the trained ANN to deduce the internal parameters of the equivalent model. All optimization algorithms utilized in this work are sourced from the Python open-source library Pymoo, a renowned repository of optimization algorithms. The algorithms within this library are highly flexible and scalable, making them suitable for addressing a diverse array of optimization challenges. To validate the efficacy and superiority of the PSO algorithm selected for parameter optimization in this paper, we also incorporated the Differential Evolution (DE) algorithm and the Non-dominated Sorting Genetic Algorithm II (NSGA2) from the Pymoo library to compare their effectiveness in the parameter optimization process.

The PSO algorithm is a heuristic search technique grounded in swarm intelligence [37,40], which addresses optimization problems by emulating the collective behaviors of birds and fish in nature. Within the PSO framework, the solution space of a problem is conceptualized as an n-dimensional domain, where each potential solution, termed a particle, possesses its own velocity and position within this domain. The algorithm progressively refines the positions and velocities of particles by considering both the individual optimal positions and the global optimal position of the entire swarm, thereby striving to discover the optimal solution. The PSO algorithm boasts advantages such as rapid convergence and high precision.

In the PSO algorithm, we configure a swarm comprising 200 particles. The number of particles is an important parameter in the PSO algorithm, as it affects the algorithm’s search capability and convergence speed. Choosing 200 particles is based on experimentation and experience. While this number may increase computational costs, having more particles can expand the coverage of the search space, increasing the probability of finding the global optimal solution. Overall, this enhances the efficiency of the algorithm. Within the solution space, characterized by 14 internal parameters of the model, the frequency required for the neural network to forecast the input is maintained within the range of 1–18 GHz, with a scanning increment of 0.1 GHz. In the update strategy of PSO, we assign the inertia factor *w* of the particles as 0.8; this is a commonly used value which dictates the extent to which the current particle’s velocity influences its velocity in the subsequent iteration. A larger inertia factor keeps particles at higher speeds, favoring global search; a smaller inertia factor decreases particle speeds, favoring local search. The value of 0.8 balances the ability for global and local search. The acceleration constants c1 and c2 are both set to 2, indicating the rates at which particles move towards their personal best (*pbest*) and the global best (*gbest*) positions. The acceleration factors (*c*1 and *c*2) represent the random acceleration weights towards the personal best (*pbest*) and global best (*gbest*) of a particle. According to Reference [41], setting the acceleration coefficients to 2 generates better solutions. It is also a compromise choice that accelerates the convergence speed of the algorithm while maintaining a certain level of stability. The formulas governing the updates for particle velocity and position are presented in Equations (Equation 3) and (Equation 4), respectively.
(3)$$v_{i}^{k+1}={wv}_{i}^{k}+c1rand_{1}{\left(0,1\right)\left(pbest_{i}^{k}-x_{i}^{k}\right)+c2rand_{2}\left(0,1\right)(gbest_{i}^{k}-x}_{i}^{k})$$
(4)$$x_{i}^{k+1}=x_{i}^{k}+v_{i}^{k+1}$$

The DE algorithm [42] is a population-based, single-objective optimization technique that iteratively refines the population through mutation, crossover, and selection operations. It is adept at solving global optimization problems involving continuous functions. The DE algorithm is distinguished by its straightforward operation, minimal parameter requirements, and ease of implementation. Initially, the DE algorithm generates mutated individuals from the initial population using a differential strategy. Subsequently, these mutated individuals are combined with target individuals according to specific rules to produce trial individuals. Finally, the fitness of the trial individuals is compared with that of the current population, and the superior individuals are retained for the next generation. This iterative process continues until the termination condition is satisfied. A pivotal parameter in the DE algorithm is the crossover constant (CR), which ranges between 0 and 1. In this experiment, we utilized the DE algorithm package from Pymoo and set the crossover constant to 0.9. This configuration enhances the interdependence of the optimization parameters and facilitates the algorithm’s convergence.

The NSGA2 [43] is an advanced genetic algorithm that seeks optimal solutions for optimization problems by emulating natural selection and genetic mechanisms. Initially, the NSGA2 algorithm performs non-dominated sorting and crowding distance calculations on the initial population. It then generates offspring through processes of selection, crossover, and mutation. These offspring are subsequently combined with the parent generation, followed by a repeat of non-dominated sorting and crowding distance calculations to select the most superior individuals to form a new parent population. This iterative process continues until the termination conditions are met. Through these steps, NSGA2 effectively identifies a set of solutions that balance multiple objectives. In this study, the implementation of NSGA2 is achieved by utilizing the default NSGA2 method available in the Pymoo library.

In selecting the objective function (fitness), we utilize the sum of the errors of the scattering parameters as the objective function, as delineated in Formula (Equation 5). In actual measurement data, the $S_{12}$ component is significantly smaller compared to other S-parameters, which can easily result in incorrect approximations when employing the PSO algorithm to fit the target. Consequently, the $S_{12}$ component in the function is multiplied by a weight coefficient of 3 to ensure the accuracy of the optimization in this part [44].
(5)$$Fitness=\Delta S_{11}+3\Delta S_{12}+\Delta S_{21}+\Delta S_{22}$$

In Formula (Equation 5), the representation of the S-parameter is shown in Formula (Equation 6).
(6)$$\Delta S_{ij}=\sum_{n=1}^{N}\left|{S_{ij,n}}^{measured}-{S_{ij,n}}^{predicted}\right|i,j=1,2;\;n=1,2\;\ldots \;N$$

The *N* in Formula (Equation 6) represents the number of data within the frequency range of 1–18 GHz.

In the algorithm comparison experiment of this study, to ensure the effectiveness of the comparison, we set the population size of the three optimization algorithms to 200 and the maximum number of iterations to 200. We verified the convergence speed and accuracy advantages of the PSO optimization algorithm in small-signal model parameter optimization through the fitness change curve of training iterations. Ultimately, the optimal solution parameters obtained by the algorithm are applied to the small-signal model and compared with the traditional optimization results derived from analytical calculations using physical formulas.

## 3. Results

The verification device utilized a China-produced DFP GaN HEMT fabricated using a 450 nm process, featuring four gate fingers, each 450 nm in length and 250 μm in width. The operational bias conditions were set to $V_{ds}$ = 28 V, 40 V, and 48 V, corresponding to $V_{gs}$ = 1 V, −1 V, and −3 V, respectively, with S-parameter measurements conducted over the range of 1–18 GHz. Under the condition of $V_{ds}$ = 28 V, $V_{gs}$ = 1 V, the gate-source voltage $V_{gs}$ was positive, and the device operated in the on-state; under the condition of $V_{ds}$ = 40 V, $V_{gs}$ = −1 V, the gate-source voltage $V_{gs}$ was negative, and the device was turned off. The device is designed for high-power applications in the S-band to Ku-band at voltage of 28 V, so an additional set of conditions with a high drain bias of $V_{ds}$ = 48 V, $V_{gs}$ = −3 V was added. To validate the proposed optimizing method for the small-signal equivalent circuit model, this paper optimized the parameters under the aforementioned three bias conditions and compared the results with those derived manually by engineers.

### 3.1. ANN Accuracy

Figure 6 illustrates the changes in test accuracy, training accuracy, and train loss during the training process of the ANN model under three distinct bias conditions. Table 2 presents the training results. The training error (Train_Loss) is computed using the MSE (Formula (Equation 1)). Both the training accuracy (Train_Accuracy) and the testing accuracy (Test_Accuracy) are determined using $R^{2}$ (Formula (Equation 2)).

Additionally, we randomly extracted data under three bias conditions beyond the training set and compared the predicted outputs of the ANN model with the simulated data. Upon validation, the comparison and error between these two sets is illustrated in Figure 7.

### 3.2. Optimization Algorithm Comparison

The trained neural network, in conjunction with the PSO algorithm, was employed to optimize the internal parameters of the small-signal equivalent circuit model under three different bias conditions. The algorithm targets the fitting of the S-parameter values of the measured device. By leveraging the neural network to rapidly predict the S-parameter performance corresponding to 15 model parameters and iterating, a set of circuit parameters suitable for the model netlist simulate is ultimately obtained. Optimization tests were carried out using thte DE, NSGA2, and PSO optimization algorithms. The trend in fitness changes during the optimization and the final fitness of the three algorithms are documented and illustrated in Figure 8.

### 3.3. ANN-PSO Optimization Results

Table 3 presents the optimization results of the model parameters using PSO under the three bias operating conditions, alongside the extraction results obtained through the conventional analytical derivation approach [22], and enumerates the absolute errors.

Moreover, the model parameters derived from PSO optimization and traditional analytical derivation are employed to simulate the equivalent circuit within the frequency range of 1–18 GHz. A comparison between the measured parameters and the simulated parameters is illustrated in Figure 9.

Additionally, we introduced the relative error of the scattering parameter [22] to assess the accuracy of the small-signal model. The error is calculated as shown in Formula (Equation 7), where *n* denotes the number of frequency points. The total percentage error $E_{TOT}$ of the model was determined by averaging the evaluated errors of the four $S_{xy}$ components of the device. The error calculation results are displayed in Table 4.
(7)$$E_{xy}{|}_{y=1,2}^{x=1,2}\;=\frac{\left|S_{xy}^{sim}\left(i\right)-S_{xy}^{meas}\left(i\right)\right|}{\sqrt{\frac{\sum_{n}{\left[S_{xy}^{meas}\right]}^{2}}{n}}}\;$$

## 4. Discussion

In the experimental results, the $R^{2}$ fitting degree of the model, as trained and tested in Table 2, exceeds 0.99. Furthermore, in the comparison of randomly selected experimental data with simulation results (Figure 7), the average error in the S-parameters remains within 0.15%. This high-precision implementation is attributed to the adoption of a rather complex neural network architecture. The magnitude of the error between the neural network predictions and the simulator simulation results is crucial for the subsequent optimization work of the algorithm. To make the neural network’s prediction accuracy approach that of the simulator, we configured 256 neurons in each hidden layer. In practice, during testing, we found that the configuration of 256 neurons exceeded the requirements, as approximately 32 neurons could achieve an accuracy of around 98%. However, this 1% difference in accuracy could lead to significant modeling errors when combined with the PSO algorithm for optimization. Therefore, we decided to increase the number of neurons to 256. With the support of modern computing resources, this configuration does not significantly increase time and computational costs during training, while also enhancing the applicability and robustness of our approach. Hence, sacrificing a bit of speed to gain this additional 1% accuracy is highly worthwhile. Additionally, we opted for ReLU as the activation function. Compared to tanh and sigmoid activation functions that require complex exponential operations, ReLU is implemented simply through a max() operation, making it computationally simpler and more cost-effective. The ReLU function also exhibits sparse activation properties, allowing some neurons in the network to remain inactive for specific inputs, thereby reducing unnecessary computations. In this way, we mitigate some of the computational burden brought by the number of neurons.

The curve in Figure 7 represents the results of our randomly selected tests, providing evidence of the success of our ANN architecture. The data in the figure is visibly fitted very well. This indicates that the model can nearly perfectly predict the outcomes for both the training and testing data, demonstrating excellent predictive performance and generalization capability. Under these circumstances, ANN models can efficiently supplant traditional simulators for the swift and accurate prediction of the simulation S-parameters of the GaN HEMT, thereby reducing simulation time and enhancing parameter tuning efficiency. For a simple ordinary circuit, a simulator typically takes several seconds to simulate once, whereas a neural network requires less than 0.001 s. During the optimization process, we continuously obtain simulation values. As the number of iterations increases, the advantages of a precise ANN model, like the one in Figure 7, in terms of speed and time become increasingly apparent, thereby enhancing overall parameter adjustment efficiency.

In the horizontal comparison experiment of the optimization algorithm, the results in Figure 8 illustrate the superiority of the PSO algorithm on the GaN HEMT small-signal model. During the optimization of the DE algorithm, there was an occurrence of premature convergence, which may be attributed to the fact that, in the optimization of the small-signal model, the value of $S_{12}$ is small and the value of $S_{21}$ is large compared to other S-parameters. Although the weights were adjusted in the fitness function, it remains challenging for the DE algorithm to significantly enhance the diversity of the particle swarm, leading to a reduction in search capability. The NSGA2 algorithm exhibits a slight deficiency in model accuracy, potentially due to its limited ability to handle multi-objective optimization problems with constraints. Compared to these two algorithms, the PSO optimization algorithm not only converges more rapidly but also demonstrates superior accuracy in solving the model parameter optimization problem presented in this paper.

The parameter optimization results of the ANN-PSO method and the traditional approach are presented in Table 3. It is evident that the optimized parameters of the neural network model are remarkably close to the outcomes derived from traditional analytic methods, indicating that the final parameters optimized by the ANN-PSO method can also be physically interpreted. Additionally, under the 40 V bias condition, it is noteworthy that the error in the value of $C_{gs2}$ is somewhat significant. According to reference [22], the $C_{gs2}$ in the classic model is 528.5. The manually tuned value was adjusted from this based on experience. The optimized $C_{gs2}$ result, in fact, is closer to the result derived from physical theory calculations, which indicates that our optimization method can reduce the errors introduced by conventional empirical adjustments to some extent.

When engineers employ traditional methods to optimize parameters, they must consider the actual physical significance to balance the relationships between the parameters of various components. The analysis and parameter adjustment work requires extensive trial and error, often taking several days to complete. In contrast, training an ANN model requires only a few hours, and it can directly yield the output results from the input parameters, without the necessity of understanding and explicating the specific relationship between the internal output values and the input values. Consequently, modelers are no longer required to engage in a complex derivation process of physical formulas to calculate parameters, thereby lowering the entry requirements for modeling optimization tasks and reducing the time cost.

In general, the integration of ANN and PSO facilitates the automation of model parameter tuning, providing significant advantages in terms of time efficiency and cost over traditional methods. Our research was conducted under three bias conditions, and the S-parameter fitting under each condition proved satisfactory. However, it is noteworthy that at $V_{ds}$ = 28 V the $S_{21}$ error reached 12.36%, compared to a manually adjusted error of 6.872%. Although the overall error remains within the 5% range, improving the *S*_21_ fitting could enhance the model’s accuracy. This discrepancy might stem from our objective function Equation 5 in the optimization algorithm, where we only considered the coefficient of the $S_{12}$ parameter. In fact, $S_{21}$ is significantly larger than others, and we also divided it by 20 when plotting the curve. Future work could involve incorporating a specific coefficient for $S_{21}$ in the objective function to improve optimization precision. From the comparative results, it is evident that the parameters optimized by the ANN-PSO method exhibit greater accuracy in fitting the $S_{11}$ and $S_{12}$ parameters of the model. Figure 9 visually depicts the S-parameter performance of the GaN HEMT small-signal model optimized by the ANN-PSO method in the Smith charts. As illustrated in Table 4, the average error of the S-parameters of the model, when compared to the measured data across the frequency range of 1–18 GHz, is below 5%. This indicates that the small-signal model optimized by ANN-PSO aligns very well with the measured data, and such an error margin is acceptable for modeling purposes.

However, we believe that there remains potential for further improvement in this accuracy. Incorporating more relevant data during training might slightly enhance the ANN model’s precision. Nevertheless, given the already high accuracy of the neural network, expanding the dataset seems unnecessary as it would also prolong the time required for parameter extraction. Our aim is to train a sufficiently precise network with minimal data. To enhance accuracy, the focus should shift towards improving the structure of the ANN or refining the search algorithms. In the future, we might consider integrating the device’s bias conditions as input parameters for neural network training. Nevertheless, this approach would complicate the creation of a high-quality dataset, as variations between curves under different conditions would become more distinct. On the other hand, this study has not significantly improved the PSO algorithm itself; future efforts could refine PSO and integrate it with ANN to boost optimization accuracy. Additionally, exploring simpler, more precise small-signal equivalent topological structures of DFP GaN HEMT models could be beneficial.

## 5. Conclusions

A rapid and efficient method has been proposed to optimize the novel small-signal equivalent circuit model of a DFP GaN HEMT. The trained neural network model achieves an impressive simulation prediction accuracy of 99.9% for the S-parameters, seamlessly supplanting the simulator. The neural network was coupled with the PSO algorithm to facilitate the automatic optimization of the equivalent circuit model parameters. The optimized small-signal model exhibits an overall average error of 4.43% when compared to the measured data of a 4 × 250 μm DFP GaN HEMT under various bias conditions, closely matching the results obtained through traditional analytical methods. Furthermore, the optimization’s impact on the performance of $S_{11}$ and $S_{22}$ surpasses that of conventional optimization techniques. Using this approach to establish a GaN HEMT model requires a short amount of time, typically just one to two hours, and the model’s accuracy is also satisfactory. This method holds promise for accelerating the equivalent modeling of various FP devices in the future.

## Figures and Tables

**Figure 1 micromachines-15-01437-f001:**
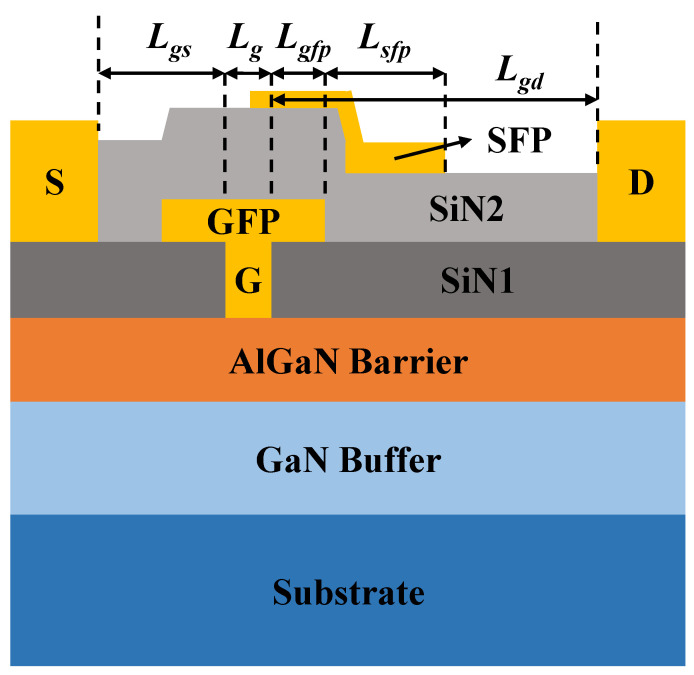
Cross-section of the DFP GaN HEMT. The letters D, S, and G indicate the drain, source, and gate, respectively. SIN2 and SIN1 represent two different silicon nitride layers.

**Figure 2 micromachines-15-01437-f002:**
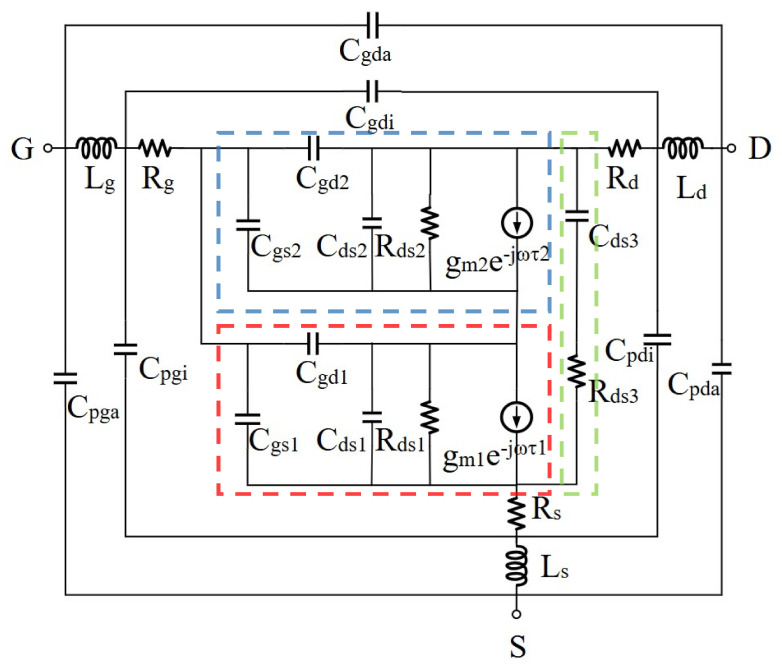
The novel complete small-signal equivalent circuit of a DFP GaN HEMT.

**Figure 3 micromachines-15-01437-f003:**
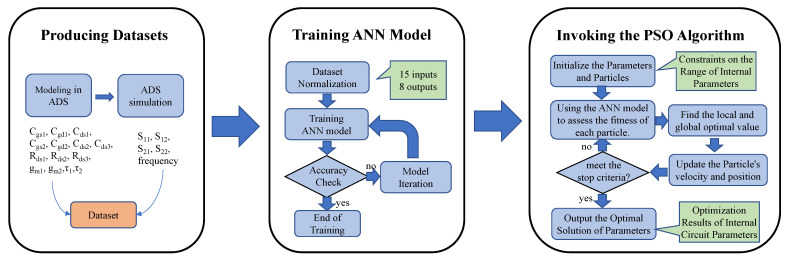
The internal parameter optimization method for the DFP equivalent circuit utilizing ANN-PSO. This method is divided into three steps: producing datasets, training the ANN model, and invoking the PSO algorithm.

**Figure 4 micromachines-15-01437-f004:**
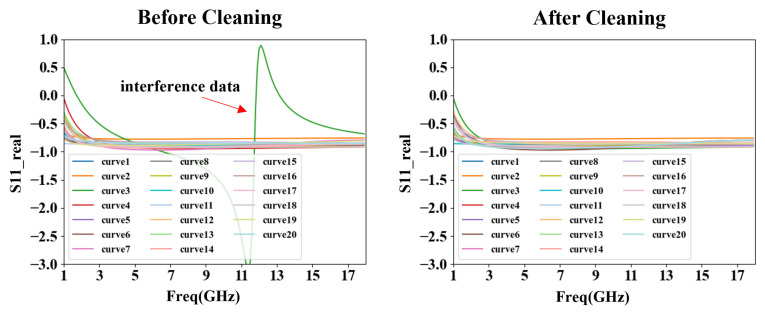
Diagram of dataset curves before and after cleaning (partial real part data of $S_{11}$).

**Figure 5 micromachines-15-01437-f005:**
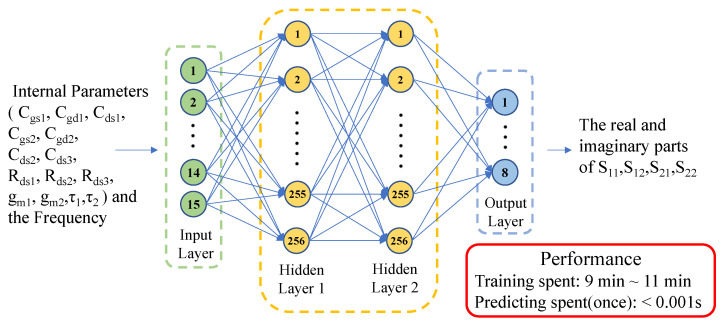
The architecture of the ANN: 1 input layer, 2 hidden layers, 1 output layer. The input layer encompasses 15 inputs, each hidden layer contains 256 neurons, and the output layer consists of 8 outputs.

**Figure 6 micromachines-15-01437-f006:**
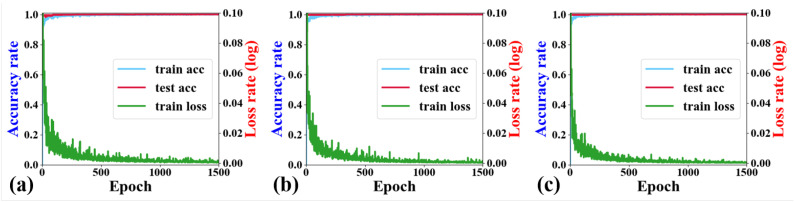
The training curves of the bias condition of (**a**) $V_{ds}$ = 28 V, $V_{gs}$ = 1 V; (**b**) $V_{ds}$ = 40 V, $V_{gs}$ = −1 V; (**c**) $V_{ds}$ = 48 V, $V_{gs}$ = −3 V. An epoch represents a complete iteration over the entire training dataset during the training process of a neural network.

**Figure 7 micromachines-15-01437-f007:**
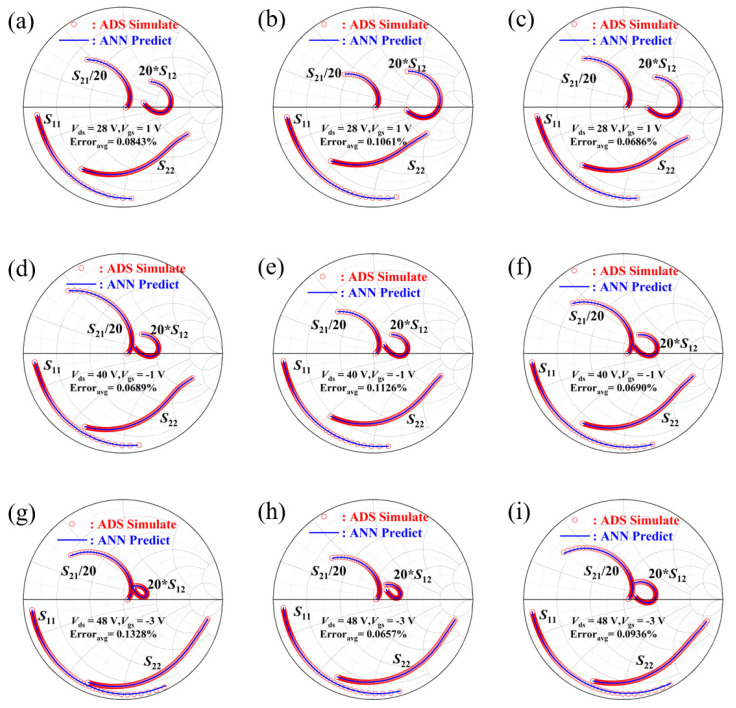
The predicted data from the model under randomly selected parameters, compared with simulated data from the simulator. (**a**–**c**) Random comparison under the condition of $V_{ds}$ = 28 V, $V_{gs}$ = 1 V; (**d**–**f**) Random comparison under the condition of $V_{ds}$ = 40 V, $V_{gs}$ = −1 V; (**g**–**i**) Random comparison under the condition of $V_{ds}$ = 48 V, $V_{gs}$ = −3 V.

**Figure 8 micromachines-15-01437-f008:**
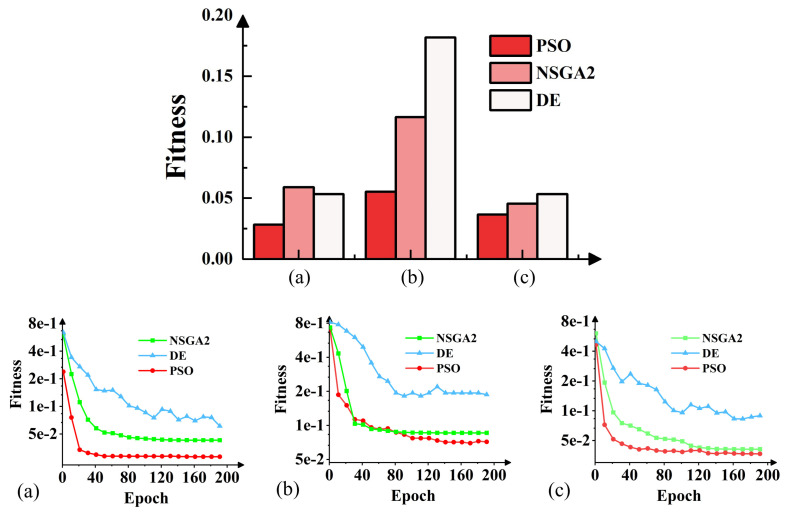
The final fitness and fitness curves of the three algorithms under bias conditions of (**a**) $V_{ds}$ = 28 V, $V_{gs}$ = 1 V; (**b**) $V_{ds}$ = 40 V, $V_{gs}$ = −1 V; (**c**) $V_{ds}$ = 48 V, $V_{gs}$ = −3 V. The epoch represents the number of iterations of the algorithm.

**Figure 9 micromachines-15-01437-f009:**
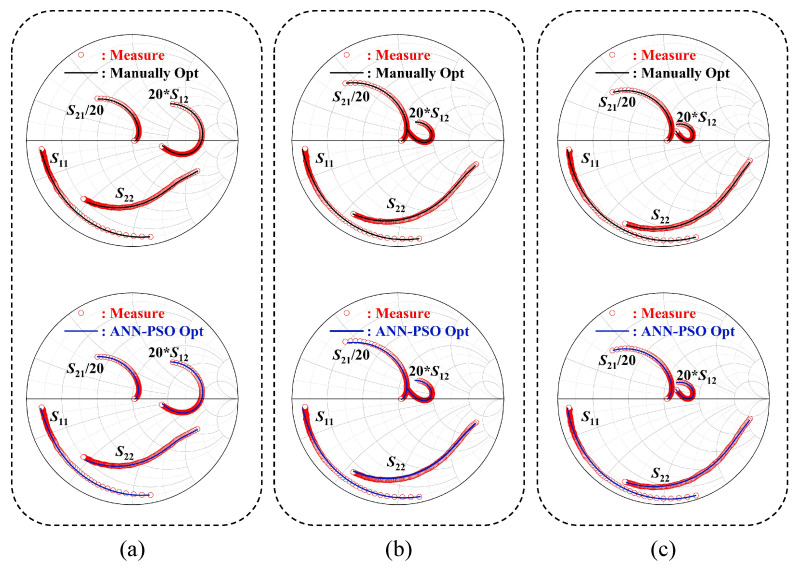
(Color online) Measured and simulated S-parameters for a 4 × 250 μm DFP GaN HEMT under the bias condition of (**a**) $V_{ds}$ = 28 V, $V_{gs}$ = 1 V; (**b**) $V_{ds}$ = 40 V, $V_{gs}$ = −1 V; (**c**) $V_{ds}$ = 48 V, $V_{gs}$ = −3 V. The frequency range is 1–18 GHz.

**Table 1 micromachines-15-01437-t001:** The maximum and minimum values of the intrinsic elements delineated for sampling.

Intrinsic Element	LB	UB
$C_{gs1}$ (fF)	500	2000
$C_{gd1}$ (fF)	200	400
$C_{ds1}$ (fF)	5	50
$C_{gs2}$ (fF)	200	700
$C_{gd2}$ (fF)	10	50
$C_{ds2}$ (fF)	1	30
$C_{ds3}$ (fF)	1	200
$R_{ds1}$ (Ω)	5	20
$R_{ds2}$ (Ω)	100	500
$R_{ds3}$ (Ω)	20	70
$g_{m1}$ (ms)	100	400
$g_{m2}$ (ms)	50	250
τ1 (ps)	3000	5500
τ2 (ps)	2500	5000

**Table 2 micromachines-15-01437-t002:** The training results of the ANN model under three bias conditions.

Bias Condition	Train_Loss (MSE)	Train_Accuracy ($R^{2}$)	Test_Accuracy ($R^{2}$)
$V_{ds}$ = 28 V, $V_{gs}$ = 1 V	0.006170	0.999919	0.999920
$V_{ds}$ = 40 V, $V_{gs}$ = −1 V	0.006420	0.999825	0.999823
$V_{ds}$ = 48 V, $V_{gs}$ = −3 V	0.005304	0.999917	0.999917

**Table 3 micromachines-15-01437-t003:** The manually tuned values and PSO optimization values of the small-signal model parameters for a 4 × 250 μm DFP GaN HEMT.

Parameter	Manually Tuned	PSO-Optimized	Absolute Error *
Bias condition at $V_{ds}$ = 28 V, $V_{gs}$ = 1 V			
$C_{gs1}$ (fF)	950.9	960.7	9.8
$C_{gd1}$ (fF)	229.0	228.7	0.3
$C_{ds1}$ (fF)	46.39	52.4	6.01
$C_{gs2}$ (fF)	338.7	359.8	21.1
$C_{gd2}$ (fF)	35.40	40.74	15.34
$C_{ds2}$ (fF)	20.31	21.93	1.62
$C_{ds3}$ (fF)	134.5	131.4	3.1
$R_{ds1}$ (Ω)	16.01	12.18	3.83
$R_{ds2}$ (Ω)	133.8	156.34	22.54
$R_{ds3}$ (Ω)	50.01	49.52	0.49
$g_{m1}$ (ms)	202.2	213.7	11.5
$g_{m2}$ (ms)	110.8	115.2	4.4
$\tau_{1}$ (ps)	4.309	4.181	0.128
$\tau_{2}$ (ps)	3.677	3.901	0.244
Bias condition at $V_{ds}$ = 40 V, $V_{gs}$ = −1 V			
$C_{gs1}$ (fF)	976.7	942.7	34.0
$C_{gd1}$ (fF)	294.8	270.2	24.6
$C_{ds1}$ (fF)	15.50	10.17	5.33
$C_{gs2}$ (fF)	660.2	414.4	245.8
$C_{gd2}$ (fF)	10.91	12.86	1.95
$C_{ds2}$ (fF)	13.66	11.05	2.61
$C_{ds3}$ (fF)	104.7	106.5	1.8
$R_{ds1}$ (Ω)	8.317	8.887	0.57
$R_{ds2}$ (Ω)	208.9	207.5	1.4
$R_{ds3}$ (Ω)	28.45	33.6	5.15
$g_{m1}$ (ms)	288.0	288.4	0.4
$g_{m2}$ (ms)	167.1	151.7	15.4
$\tau_{1}$ (ps)	4.307	4.160	0.147
$\tau_{2}$ (ps)	3.624	3.931	0.307
Bias condition at $V_{ds}$ = 48 V, $V_{gs}$ = −3 V			
$C_{gs1}$ (fF)	913.1	898.6	14.5
$C_{gd1}$ (fF)	244.3	264.5	20.2
$C_{ds1}$ (fF)	9.731	8.761	0.97
$C_{gs2}$ (fF)	539.1	459.7	79.4
$C_{gd2}$ (fF)	13.13	12.94	0.19
$C_{ds2}$ (fF)	23.39	23.58	0.19
$C_{ds3}$ (fF)	60.77	62.03	1.26
$R_{ds1}$ (Ω)	6.449	6.784	0.335
$R_{ds2}$ (Ω)	351.8	340.1	11.7
$R_{ds3}$ (Ω)	44.1	45.4	1.3
$g_{m1}$ (ms)	207.6	223.9	16.3
$g_{m2}$ (ms)	179.3	164.3	15
$\tau_{1}$ (ps)	4.609	4.002	0.607
$\tau_{2}$ (ps)	3.881	3.892	0.011

* Absolute error: The absolute value of the difference between two sets.

**Table 4 micromachines-15-01437-t004:** The error comparison between the proposed method and the traditional approach under three distinct bias conditions.

Bias Condition	Optimization Method	$E_{11}$ (%)	$E_{12}$ (%)	$E_{21}$ (%)	$E_{22}$ (%)	$E_{\mathrm{TOT}}$ (%)
$V_{ds}$ = 28 V	Manual	1.549	3.192	6.872	2.393	3.5015
$V_{gs}$ = 1 V	ANN-PSO	1.231	3.199	12.36	2.791	4.8953
$V_{ds}$ = 40 V	Manual	1.176	4.797	4.072	2.173	3.0545
$V_{gs}$ = −1 V	ANN-PSO	1.637	5.369	5.391	3.790	4.0468
$V_{ds}$ = 48 V	Manual	1.314	9.284	4.511	2.576	4.4213
$V_{gs}$ = −3 V	ANN-PSO	1.007	9.248	5.015	2.172	4.3605

## Data Availability

The original contributions presented in the study are included in the article, further inquires can be directed to the corresponding authors.

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
