# Peer review of "A Novel ANN-PSO Method for Optimizing a Small-Signal Equivalent Model of a Dual-Field-Plate GaN HEMT"

_micromachines, 2024, doi:10.3390/mi15121437_

Round 1

Reviewer 1 Report

Comments and Suggestions for Authors

Dear authors,

I found your manuscript a well-written, detailed paper about your research. This paper explores an ANN-PSO approach that replaces traditional simulation and manual parameter tuning with automated, efficient model optimization, effectively demonstrates the benefits of using PSO method for optimizing the small-signal equivalent model of dual field-plate GaN HEMT.

I have a few comments listed below:

1)            Since you have used FP for field-plate on line 18-19, why not use it throughout the manuscript? That would simplify the wording and make reading easier. You can also use DFP for dual field plate since you used SFP and GFP on line 24.

2)            Line 39: Propose, not proposes.

3)            Explain the letters like S, G, D, etc. in Fig. 1 in the captions. People who are not familiar with HEMT will not understand this figure very well.

4)            Add reference for Fig. 2.

5)            Line 102, you need to define/introduce ADS.

6)            Table 1, please use lower case s for second.

7)            Font size is a bit small for Fig. 4. Also, there’s not much explanation of what Fig. 4 is showing either in the main text or the caption. Please add that information to the manuscript.

8)            Line 160, you don’t need to write out PSO again since it was introduced already. Similar comments can apply to other places as well.

9)            Line 223, fix the typo of the unit of width.

10)        How about merging Fig. 8 and Fig. 9 since their captions are almost the same.

11)        I don’t like the way Table 3 presents data. Please change the format so each parameter is individually listed across different columns. Also, you said it’s evident that the optimized parameters are remarkably close to the traditional analytic methods. That’s a very strong claim, but it’s not clear at all from table 3. Especially for cg2 values, the errors are huge. Please further explain it so your claim is supported.

12)        In conclusion, you said it’s a rapid method. But you have not mentioned anywhere about the cost in time of your optimization method. You would need to add that information to the main text to support it.

13)        The average error of below 5% is not bad but also not too impressive. Could you briefly discuss how to further improve the error rate in future studies?

Reviewer 2 Report

Comments and Suggestions for Authors

In the manuscript, a new approach is proposed to optimize the small-signal equivalent circuit model of GaN HEMT device, in which ANN and PSO algorithms are combined. There are some merits in the manuscript. However, following issues should be addressed:

1. Introduction

More backgraound analysis on the published optimization approaches for the model's parameters should be added.

2. Materials and Methods

2.1. Novel Device Model

Why the 450-nm device is adopted in the manuscript? Please show the feasibility of spreading the approach into other devices, with the same material but different technology. 

2.2 Model Optimization Method

2.2.1. Producing Datasets

The producing process of datasets is introduced in the part. Please explain why three kinds of bias condictions are used. In Table.1, the minimum and maximum values are listed. What's the reference standard for the range? For the data exclusion, how to choose the interference data? Why the resonant data in specific bias condition (e.g., Vds=28V, Vgs=1V) is executed the exclusion? 

2.2.3. Invoking the PSO algorithm

More detail explanation on how to use the predicted S parameters by ANN for PSO alglrithm. Why choose 200 particles, intertia factor 0.8 and acceleration factor 2? Comparison should be made to show the influence of different parameters on the results.

3. Results

Discussion on the results of Figure 7 should be conducted in more detail.

4. Discussion

The research in the paper is based on the training on three bias conditions. How about the accuracy of the adopted three bias conditions? Authors should explain the discrepency if other bias conditions are applied.Also, if more applied data are included in the training process, can the accuracy be improved? 

Reviewer 3 Report

Comments and Suggestions for Authors

The authors propose a method to enhance the automation to small signal equivalent model extraction of a dual field-plate GaN HEMT. However, the author’s presentation of the overall manuscript is not enough clear to quantitative and qualitative highlight the methods guided with a clear methodology for the circuit modeling and simulation to demonstrate characteristics parameters for the simulation results Figures 2, and Table 1.

Additionally, in Figure 5 ANN configurations do not provide consistently observations and systematic results for parameters such as time convergence, and complexity to evaluate a qualitative significance of the contribution in the method automation to affirmative as simple yet effective for a practical application or automation of the model parameter tuning.

Figure 7 shows simulated and predicted data with a good match, however the proposed curves do not show a complex model extraction/prediction to fit the provided dataset to justify the high complexity in the architecture proposed with 256 neurons as depicted in Fig. 5. Authors are encouraged to provide a more detailed explanation of the model error performance, selection of the model parameters for the ANN ReLU architecture parameterization and fitting results

Also, it is not clear if the measured data such as S-parameters are directly extracted from the dual-field-plate GaN HEMT fabricated using a 450 nm process, a testbench used to characterization and measurements extraction can provide more clarity to provide a confidence on the key contribution of this study; therefore, a testbench description is necessary to highlight the extraction measurements procedure  for example to give the possibility to scaling the metrics under another extrinsic characteristics for another gate lengths.

Comments on the Quality of English Language

English could be improved to more clearly express the research, specifically in grammar.

Round 2

Reviewer 2 Report

Comments and Suggestions for Authors

I am glad to see my concerns have been addressed.